# PTRF/Cavin-1 promotes efficient ribosomal RNA transcription in response to metabolic challenges

**Libin Liu[1]\*, Paul F Pilch[1,2]\***

[1]Department of Biochemistry, Boston University School of Medicine, Boston, United States; [2]Department of Medicine, Boston University School of Medicine, Boston, United States

**Abstract** Ribosomal RNA transcription mediated by RNA polymerase I represents the rate-limiting step in ribosome biogenesis. In eukaryotic cells, nutrients and growth factors regulate ribosomal RNA transcription through various key factors coupled to cell growth. We show here in mature adipocytes, ribosomal transcription can be acutely regulated in response to metabolic challenges. This acute response is mediated by PTRF (polymerase I transcription and release factor, also known as cavin-1), which has previously been shown to play a critical role in caveolae formation. The caveolae–independent rDNA transcriptional role of PTRF not only explains the lipodystrophy phenotype observed in PTRF deficient mice and humans, but also highlights its crucial physiological role in maintaining adipocyte allostasis. Multiple post-translational modifications of PTRF provide mechanistic bases for its regulation. The role of PTRF in ribosomal transcriptional efficiency is likely relevant to many additional physiological situations of cell growth and organismal metabolism.

**\*For correspondence:** libin@bu. edu (LL); ppilch@bu.edu (PFP)

**Competing interests:** The authors declare that no competing interests exist.

## Introduction

The ribosome is the engine for protein synthesis and it is vital for cell growth and survival. Ribosome biogenesis, the process of making ribosomes, is a complex and energy intensive process that involves transcriptional regulators, protein modification, and the assembly of multi-component macromolecular complexes (reviewed in (*Grummt, 2003*; *Kusnadi et al., 2015*; *Russell and Zomerdijk, 2005*). Synthesis of ribosomal RNA (rRNA) by RNA polymerase I (Pol I) is generally considered as a major rate-limiting step in ribosome biogenesis, and it produces the common 47S precursor of mature 5.8S, 18S, and 28S rRNAs (review in [*Moss et al., 2007*]). The 47S pre-rRNA is transcribed from hundreds of copies of ribosomal DNA (rDNA) genes distributed in repeated arrays among 5 acrocentric chromosomes in humans (*Henderson et al., 1972*; *Stults et al., 2008*). Given the large numbers of rDNA genes, rRNA transcription can be regulated by both varying the number of transcriptionally active genes and/or by varying the rate of activated transcription. The former can be regulated through chromatin remodeling and related factors, and the latter can be acutely regulated through the factors assembled in Pol-I transcription complex machinery (reviewed in (*Grummt, 2010*).

Ribosomal transcriptional activity has been tightly linked to cell growth and proliferation. Any perturbation that changes cell growth or protein synthesis, such as nutrient and growth factor availability, senescence, toxin exposure or viral infection, leads to changes in rDNA transcription activity (reviewed in (*Kusnadi et al., 2015*). This regulatory process comprises a series of coordinated steps including transcription initiation, promoter escape, elongation and termination (reviewed in (*Russell and Zomerdijk, 2005*; *Schneider, 2012*). Regulation is achieved *via* coordinated multiple

**eLife digest** Obesity can cause several other health conditions to develop. Type 2 diabetes is one such condition, which arises in part because fat cells become unable to store excess fats. This makes certain tissues in the body less sensitive to the hormone insulin, and so the individual is less able to adapt to changing nutrient levels. Without treatment or a change in lifestyle, this insulin resistance may develop into diabetes. However, "healthy obese" individuals also exist, who can accommodate an overabundance of fat without developing insulin resistance and diabetes.

Some forms of rare genetic disorders called lipodystrophies, which result in an almost complete lack of body fat, can also lead to type 2 diabetes. This raises the question of whether lipodystrophy and obesity share some common mechanisms that cause fat cells to trigger insulin resistance. One possible player in such mechanisms is a protein called PTRF. In rare cases, individuals with lipodystrophy lack this protein, and mice that have been engineered to lack PTRF also largely lack body fat and develop insulin resistance.

Fat cells can respond rapidly to changes in nutrients during feeding or fasting, and to do so, they must produce new proteins. Structures called ribosomes, which are made up of proteins and ribosomal RNA, build proteins; thus when the cell needs to make new proteins, it also has to produce more ribosomes. PTRF is thought to play a role in ribosome production, but it is not clear how it does so.

Liu and Pilch analyzed normal mice as well as those that lacked the PTRF protein. This revealed that in response to cycles of fasting and feeding, PTRF increases the production of ribosomal RNA in fat cells, enabling the cells to produce more proteins. By contrast, the fat cells of mice that lack PTRF have much lower levels of ribosomal RNA and proteins.

Liu and Pilch then examined mouse fat cells that were grown in the laboratory. Exposing these cells to insulin caused phosphate groups to be attached to the PTRF proteins inside the cells. This modification caused PTRF to move into the cell's nucleus, where it increased the production of ribosomal RNA.

Overall, the results show that fat cells that lack PTRF are unable to produce the proteins that they need to deal with changing nutrient levels, leading to an increased likelihood of diabetes. The next steps are to investigate the mechanism by which PTRF is modified, and to see whether the mechanisms uncovered in this study also apply to humans.

signaling pathways that modulate the expression and activities of many key factors, such as Pol I-specific transcription factors (RRN3, also known as transcription initiation factor 1A, TIF-1A), selectivity factor 1 (SL-1; also known as TIF-1B), upstream binding factor (UBF) and others (reviewed in (*Bywater et al., 2013*; *Grummt, 2010*). Most of these studies were performed in cultured cancer or cancer-like cell lines, where ribosomal transcriptional regulation was coupled to cell proliferation or closely related cell growth. For cell mass growth in mature non-proliferating cells, it's less clear if and how ribosomal transcription is regulated, and the physiological relevance of ribosomal RNA transcription in these cells has been little studied.

Adipocytes are a highly metabolically dynamic cell type that can respond rapidly to alterations in nutrient excess and deprivation, thereby fulfilling its major role in whole body energy homeostasis (reviewed in (*Rosen and Spiegelman, 2014*; *Scherer, 2006*; *Sun et al., 2011*). As a mature non-proliferating cell type, it undergoes dramatic changes upon metabolic challenges. In obesity due to excess calorie loading, adipocytes need to develop not only corresponding cellular structures and functions for the increasing needs in lipid storage and metabolic capacity, but also the machinery for the secretion of adipokines and other proteins. These cells also have to maintain an insulin sensitive functional response in order to avoid the development of type 2 diabetes. Given the importance of homeostatic protein synthesis as a basic cellular function to maintain structure and activity, and to ensure normal cellular physiological functions, it becomes obvious that 'healthy' adipocyte expansion has to be supported by fundamental processes such as protein synthesis, which in turn, can be determined by ribosome biogenesis. Changes in ribosomal RNA synthesis by long term starvation and re-feeding were in fact reported soon after ribosomes were first described (*Benjamin and*

*Gellhorn, 1966*) although many mechanistic details of ribosome composition and function were unknown at that time. The effect of insulin on protein synthesis and ribosome biogenesis in adipocytes was also reported (*Hansson and Ingelman-Sundberg, 1985*; *Vydelingum et al., 1983*). A precisely controlled ribosomal DNA transcriptional response to changes in nutrient and insulin levels would therefore seem essential for healthy adipocytes.

We and others have shown that PTRF (polymerase I transcription and release factor, also known as Cavin-1, herein after, PTRF), plays a critical role in caveolae formation (*Hill et al., 2008*; *Liu et al., 2008*; *Liu and Pilch, 2008*), structures that are particularly abundant in adipocytes. Moreover, a lipodystrophic phenotype was observed in PTRF null mice (*Ding et al., 2014*; *Liu et al., 2008*) that is similar or identical to that of human patients with inactivating PTRF mutations who also display a type of muscular dystrophy (*Ardissone et al., 2013*; *Dwianingsih et al., 2010*; *Hayashi et al., 2009*; *Jelani et al., 2015*; *Shastry et al., 2010*). The molecular mechanisms underlying these phenotypes that have been proposed, principally alterations in lipid metabolism/transport and perturbations of the cell surface membrane (*Parton and del Pozo, 2013*; *Pilch and Liu, 2011*) cannot fully explain both the adipose and muscular dystrophy phenotypes.

In fact PTRF/Cavin-1 as PTRF was first characterized by its Pol-I related regulatory function (*Jansa et al., 1998*, *2001*; *Jansa and Grummt, 1999*). These in vitro studies established a role for PTRF in the efficiency of rRNA transcription (*Jansa et al., 1998*, *2001*; *Jansa and Grummt, 1999*), but since then no further experiments concerning this function have been performed that we are aware of. Moreover, the physiological relevance of this activity was never established in cells or in vivo. Consequently, we used primary mouse and cultured adipocyte experimental systems to show that PTRF localized to the nucleus and associated with the pol I transcription complex, playing a direct role on metabolically regulated ribosomal DNA transcription. A number of PTRF post-translational modifications and motifs can explain its nuclear translocation and the role in mediating insulin and nutrient-regulated ribosomal transcription. PTRF also plays a critical role in maintaining the active rDNA transcriptional 'loop' formation. Our studies not only show a specific role of PTRF on metabolism-regulated ribosomal DNA transcription in the adipocyte, they add another layer of regulation to rDNA transcriptional complexity. They also highlight the role of ribosome biogenesis in adipocyte allostasis.

## Results

### Insulin stimulates PTRF nuclear translocation to the nucleus and binding to TTF1 at the ribosomal DNA transcription termination region

Using immunofluorescence staining, we showed that PTRF has a clear insulin stimulated nuclear translocation in cultured 3T3-L1 adipocytes (*Figure 1A*). After insulin stimulation, PTRF but not caveolin-1 showed significant nuclear localization in isolated primary (Top panels) and cultured 3T3-L1 adipocytes (Bottom panels) (*Figure 1B*). To further confirm this, we fractionated 3T3-L1 adipocytes into nuclear and non-nuclear fractions by sucrose gradient centrifugation, and as shown in *Figure 1C*, a significant portion of PTRF translocated to nucleus upon insulin stimulation. In contrast other caveolae component proteins such as caveolin-1, Cavin-2 (SDPR), and Cavin-3 (SRBC) did not show any nuclear localization or stimulated translocation (*Figure 1C*), suggesting a specific role of PTRF on rRNA transcriptional regulation. We performed an immunoprecipitation from the nuclear fraction with PTRF antibody, and consistent with the prior in-vitro assay (*Jansa et al., 1998*), our data indicated PTRF binds to TTF1 (polymerase I transcription termination factor 1) in cells (*Figure 1D*). We also performed chromatin immunoprecipitation using a PTRF antibody followed by qPCR using primers probing the ribosomal transcription termination region. As shown in *Figure 1E*, the occupancy of PTRF in the termination complex is significantly regulated by the nutrient availability and insulin. These results show PTRF nuclear localization and binding to the rDNA transcription complex, suggesting that PTRF might play a functional role in ribosomal transcription activity in cells.

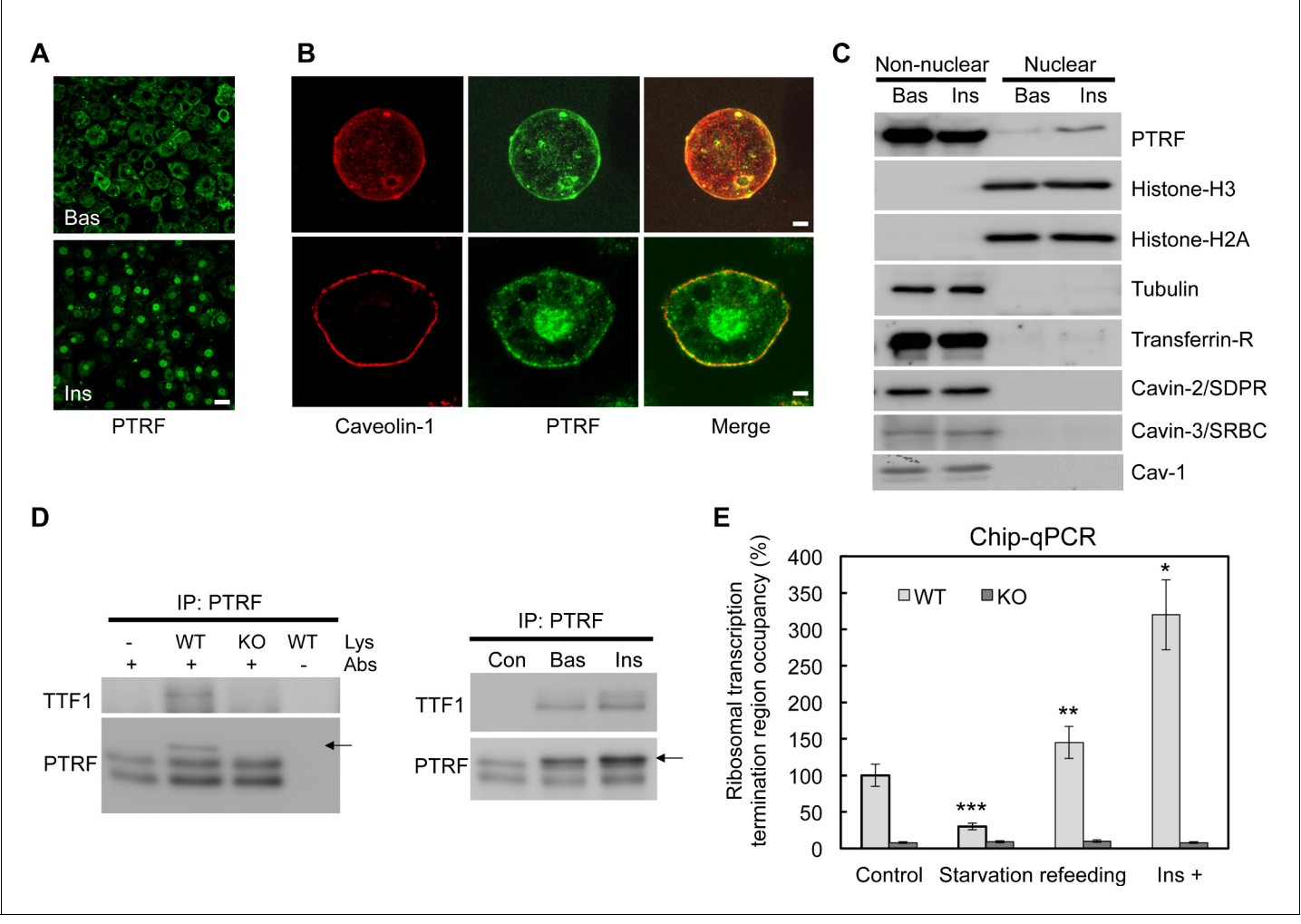

**Figure 1.** A functional role of PTRF in nucleus. (**A**) Immunofluorescence staining of PTRF in basal and insulin stimulated 3T3-L1 adipocytes. Scale bars, 50 µm. (**B**) Higher magnificent images of insulin stimulated single isolated primary (top) and 3T3-L1 adipocyte (bottom) are shown, green: PTRF, and red: caveolin-1. Scale bars, 10 µm. (**C**) Nuclear and non-nuclear fractions from basal or stimulated primary adipocytes were subject to western blots by indicated antibodies. (**D**) Co-immunoprecipitation by PTRF antibody from basal and stimulated adipocytes nuclear lysate followed by western blots using TTF1 (Transcription Termination Factor, RNA Polymerase I) and PTRF antibodies, PTRF null adipocytes (KO) serving as a negative control for the co-IP. (**E**) Chip-qPCR assay using PTRF antibody from wild type control (WT) and PTRF null adipocytes (KO). *p<0.05, **p<0.01, and ***p<0.001; Student's test. Error bars indicate SD.

## Ribosomal transcription in adipocytes is regulated by metabolic challenges and loss of PTRF causes its dysregulation

Previous studies have shown that rDNA transcription can be acutely regulated by nutrients and growth factors (reviewed in [*Moss et al., 2007*]). After pre ribosomal RNA (pre-rRNA) is synthesized, the first step is to remove the 5' portion of the 5'-externally transcribed spacer (5'-ETS) to generate 45S pre-rRNA (*Craig et al., 1987*; *Gurney, 1985*; *Miller and Sollner-Webb, 1981*). The length of the removed region, known as the leader fragment of the 5'-ETS, is 650 bases in mouse and 414 bases in human. Because this leader sequence is rapidly excised and degraded after transcription (*Puvion-Dutilleul et al., 1997*), its quantity in the nucleolus is a sensitive and highly selective marker of the amount of the nascent 47S pre-rRNA and hence, a widely used indicator of the rate of ribosomal transcription. Here we let 3T3-L1 fibroblasts and differentiated adipocytes 'rest' in PBS with 1% BSA for 3–4 hr, then switched them back to full culture medium (10% FBS in high glucose DMEM) with insulin. As shown in *Figure 2A*, pre-rRNA 47S levels were significantly up-regulated by

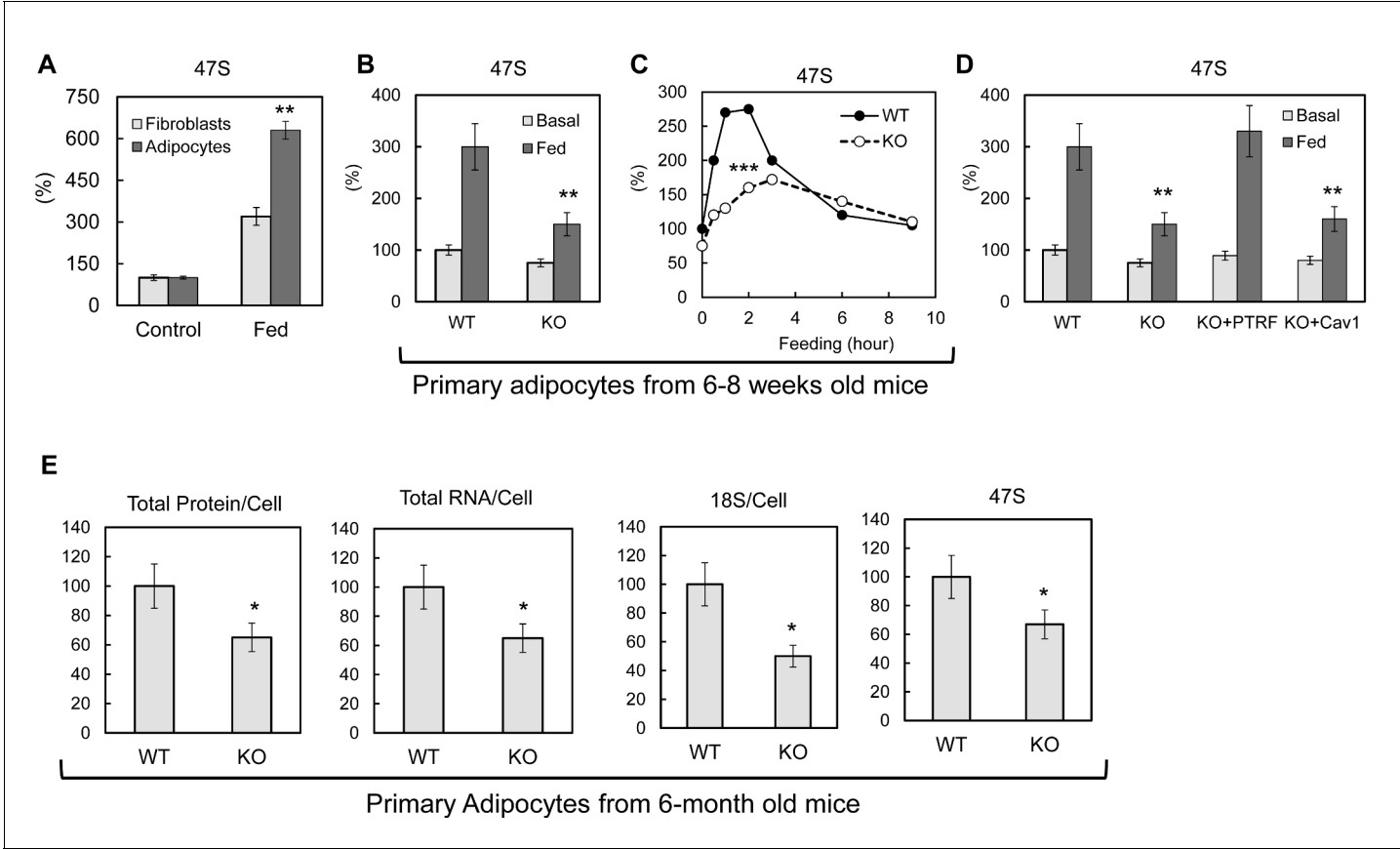

**Figure 2.** PTRF mediated ribosomal transcription regulations. (**A**) Relative pre-rRNA (47S) levels were detected by RT-qPCR from 3T3-L1 fibroblast and fully differentiated adipocytes stimulated by switching culture medium from 1% BSA in PBS to full growth medium with insulin (Fed). (**B**) Basal (KRP buffer with 1% BSA) or 45 min stimulated (Fed: high glucose DMEM, 10% FBS, and insulin) pre-rRNA levels were detected by RT-qPCR in primary adipocytes isolated from wild type (WT) and PTRF null (KO) mice. (**C**) A stimulation time-course of B. (**D**) Pre-rRNA levels were detected by RT-qPCR from basal or stimulated wild type (WT), PTRF null (KO), PTRF re-transfected PTRF null (KO+PTRF), and Cav-1 re-transfected PTRF null (KO+Cav1) MEFs cells. (**E**) Relative levels of total proteins, RNA, 18S, and 47S from wild type or PTRF null mice (n = 6) were measured and normalized to cell numbers. *p<0.05, **p<0.01, and ***p<0.001; Student's test. Error bars indicate SD.

The following figure supplements are available for figure 2:

**Figure supplement 1.** Characterization of ribosome biogenesis in 6–8 weeks old PTRF null and control mice.

**Figure supplement 2.** Cell growth curve of cultured WT and PTRF null MEFs cells.

this culture condition change. Compared to fibroblastic cells, the response in adipocytes is 2-fold higher, suggesting an adipocyte-specific rDNA transcription regulation mechanism as the levels of PTRF are much higher in the latter as compared to the former (*Bastiani et al., 2009*). To see if PTRF played any role on this regulation, we isolated primary adipocytes from 6–8 weeks old wild type and PTRF null mice and challenged them with similar medium changes by switching from 1% BSA/KRP buffer to 10% FBS in high glucose DMEM with 100nM insulin. A dramatic defect of rDNA transcription up-regulation was observed in PTRF null cells as compared to wild type cells (*Figure 2B*). A time-course experiment further showed the peak of rDNA transcription in response to a metabolic challenge was significantly blunted in PTRF null cells (*Figure 2C*).

To further confirm this is a PTRF-dependent effect, similar experiments were performed in cultured primary (within 2–3 passages) wild type, PTRF null, and PTRF or caveolae-1 rescued null MEF cells. To exclude cell proliferation associated ribosomal transcription, cells were pre-treated with nocodazole to induce cell arrest by depolymerizing the microtubule network, and culture media

were switched from 1% BSA in PBS to full culture medium with insulin. Consistent with the results from adipocytes, PTRF null MEFs showed a dramatically smaller response to the culture condition change. PTRF transfection rescued this defect, whereas caveolin-1 (Cav-1) cannot, indicating a role of PTRF independent of its function in caveolae (*Figure 2D*).

To determine if loss of this acute PTRF-dependent rDNA transcriptional regulation caused any long term ribosome biogenesis defect, we measured total protein, RNA and pre-rRNA contents in primary adipocytes from PTRF null and control wild type mice. Although 6–8 week old mice did not show any differences (*Figure 2—figure supplement 1*), in 6-month old mice, total protein and RNA were significantly reduced in PTRF null primary adipocytes (*Figure 2E*). A significantly lower basal level of 47S was also observed. In addition, PTRF null MEFs showed a slower growth rate comparing to wild type control (*Figure 2—figure supplement 2*). These data support a ribosome degeneration phenomenon, which may explain the in vivo dystrophic phenotype we observed in PTRF null mice (*Ding et al., 2014*; *Liu et al., 2008*), also seen in human patients with PTRF mutations (*Hayashi et al., 2009*).

## Loss of PTRF-dependent ribosomal transcriptional regulation is the direct causal mechanism for the pathogenesis of lipodystrophy

To determine if PTRF played a direct causal effect on rDNA transcription, we created a PTRF null stable 3T3-L1 cell line using CRISPR/cas9 genome editing technology. Three different *Ptrf* gene exon loci were targeted. Similar results were obtained from all three cell-lines (*Figure 3—figure supplement 1A*) and results from one of these (KO3) are shown here (*Figure 3A*). From the cellular lipid content and adiponectin (Adpn) secretion (*Figure 3B*) and the expression of key adipocyte differentiation markers of 3T3-L1 by western blot (*Figure 3C*) and qPCR (*Figure 3—figure supplement 1B*), we concluded that the PTRF null 3T3-L1 cells do not have any significant deficit in the degree of 3T3-L1 adipocyte differentiation as compared to control cells. We also did not see any significant changes for total protein, RNA and 47S rRNA levels (*Figure 3—figure supplement 1C*). However when we subject the cells to metabolic challenges by switching culture medium from PBS with 1% BSA ('fasting') to high glucose DMEM with 10% FBS and insulin ('feeding'), 47S levels in PTRF null adipocytes were not up-regulated as efficiently as in control cells (*Figure 3D*). A further time-course study showed a delayed response upon loss of PTRF. Conversely, 47S levels in PTRF null adipocyte were not down-regulated as efficiently as in wild type cells under nutritional challenge by switching from full culture medium to PBS with 1% BSA (fasting). These data further support a crucial functional role of PTRF on metabolically regulated rDNA transcription.

Unlike the cell culture model, humans and mice are constantly challenged by daily feeding or fasting cycles. To mimic these physiological conditions in vitro, we repeatedly challenged WT and PTRF null 3T3-L1 adipocytes by repeated 'feeding', and 'fasting' at 12-hr intervals. After seven days, the PTRF null adipocytes showed a dramatically lower lipid content, less total protein and less total and 47S RNA levels (*Figure 3E*), which essentially recapitulated the lipodystrophy characteristics observed in the PTRF knockout mouse model and human patients with PTRF deficiencies. These data suggest a PTRF-dependent ribosomal transcriptional response may be the early and direct causal mechanism for the pathogenesis of in vivo lipodystrophy. Interestingly we saw a subunit of RNA polymerase I, PolR1A dramatically decreased upon 3T3-L1 differentiation, whereas PTRF and TTF1 levels increased (*Figure 3C*), suggesting a possible adipocyte-specific termination factor-dependent mechanism for the rDNA transcription regulations (See Discussion).

## The p53 pathway is dysregulated upon loss of PTRF

It has been reported that deregulated ribosomal biogenesis can induce p53 pathway activation (*Woods et al., 2015*; *Zhang and Lu, 2009*), which will lead to cell growth arrest, apoptosis and cell death (*Berkers et al., 2013*; *Vousden and Prives, 2009*). Here we first examined p53 activation in PTRF null 3T3-L1 adipocytes. We did not observe a significant p53 and MDM2 protein level change at basal condition (*Figure 4B*, left). However when we challenged the cells with 'feeding' medium, PTRF null cells showed a poor rRNA response (*Figure 4A*), higher ribosomal protein rpl5 levels, lower mdm2 and higher p53 nuclear accumulation (*Figure 4B*, right), indicating p53 pathway activation. This is probably due to insufficient rRNA synthesis causing a relative increased accumulation of ribosomal proteins in the nucleus, which is also called nuclear stress. When we challenged the cells

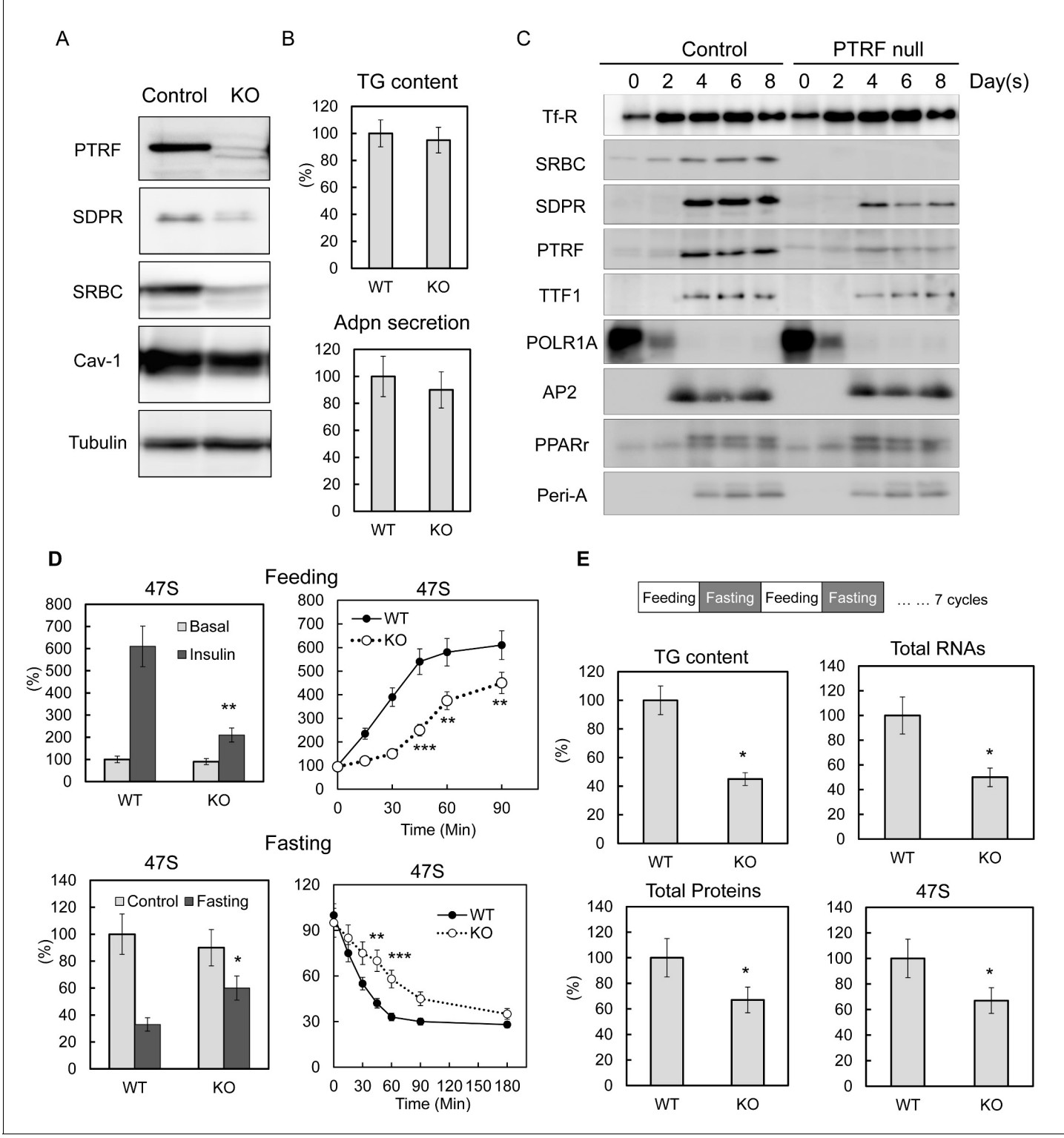

**Figure 3.** PTRF plays critical role mediating ribosomal transcription response to metabolic challenges in CRISPR/Cas9 genomic editing cell model, without any affect on differentiation. (**A**) Indicated protein levels were detected by Western blots from CRISPR/Cas9 genomic edited PTRF null and control 3T3-L1 adipocytes. (**B**) Relative total TG content and adiponectin secretion levels were measured from CRISPR/Cas9 genomic edited PTRF null and control 3T3-L1 adipocytes. (**C**) Expression levels changes of indicated proteins during 3T3-L1 differentiation were measured by western blot from CRISPR/Cas9 genomic edited PTRF null and control 3T3-L1 adipocytes. (**D**) Pre-rRNA levels from nutrients/insulin stimulated (feeding: switching from PBS with 1% BSA to high-glucose DMEM with 10% FBS and insulin) or starved (Fasting: switching the growth medium to PBS with 1% BSA) were

*Figure 3 continued on next page*

*Figure 3 continued*

measure by RT-qPCR from CRISPR/Cas9 genomic edited PTRF null and control 3T3-L1 adipocytes. (E) After CRISPR/Cas9 genomic edited PTRF null and control 3T3-L1 adipocytes were subject to 'feeding' 12 hr followed by 'fasting' 12 hr cycle for seven days, total TG content, proteins, RNAs and 47S levels were measure as described before. *p<0.05, **p<0.01, and ***p<0.001; Student's test. Error bars indicate SD.

The following figure supplement is available for figure 3:

**Figure supplement 1.** Characterizations of CRISPR/Cas9 genomic edited PTRF null 3T3-L1 adipocytes.

by 'fasting' medium, the inefficient rRNA transcription regulation was confirmed in PTRF null cells comparing to WT (*Figure 4C*). However we saw opposite responses of p53 activation, which was transiently up-regulated in control wild type cells and lost in PTRF null adipocytes (*Figure 4D*). This response in wild type cells seems to be only at cell arrest levels (activated *p21*, *gadd45*, and *cdkn1a*), not cell apoptosis (unchanged levels of *atm*, *Bax* and *caps2*) (*Figure 4E*), which probably represents some level of cell response under nutrient starved conditions. When we examined the adipose tissue from PTRF null mice, a full range of p53 activation targets including both cell arrest and apoptosis all were up regulated (*Figure 4F–G*), indicating a long term adipocyte apoptosis.

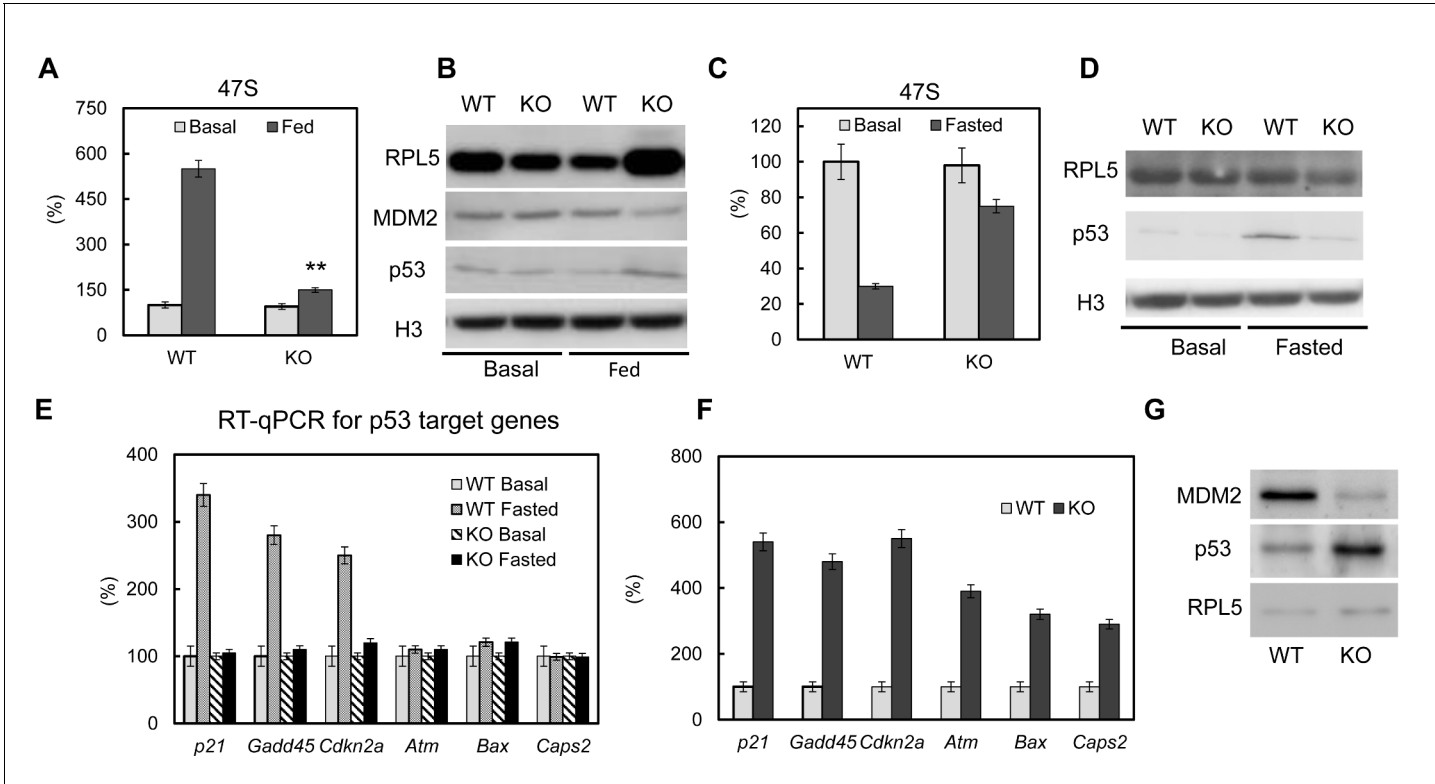

**Figure 4.** The p53 pathway is dys-regulated upon loss of PTRF. Pre-rRNA levels from fed (A) and fasted (C) control and CRISPR/Cas9 edited PTRF null adipocytes were determined by RT-qPCR. Nucleus fraction lysates from fed (B) or fasted (D) adipocytes were separated in SDS-page followed by western blot by using the indicated antibodies, RPL5: ribosomal protein L5; MDM2: mouse double minute 2 homolog; H3: histone-H3. (E) p53 pathway related marker gene expression levels were measured from basal and fasted control or CRISPR/Cas9 edited PTRF null adipocytes by RT-qPCR. *Gadd45: Growth arrest and DNA-damage-inducible protein GADD45; Cdkn2a: cyclin-dependent kinase inhibitor 2; Atm: Ataxia telangiectasia mutated; Bax: bcl-2-like protein 4; Caps2: Calcyphosine 2.* (F) p53 pathway related marker gene expression levels were measured from six-month old wild type control and PTRF null mice adipose tissues. (G) Wild type and PTRF null mice adipose tissue lysate were subject to western blots by using indicated antibodies. **p<0.01; Student's test. Error bars indicate SD.

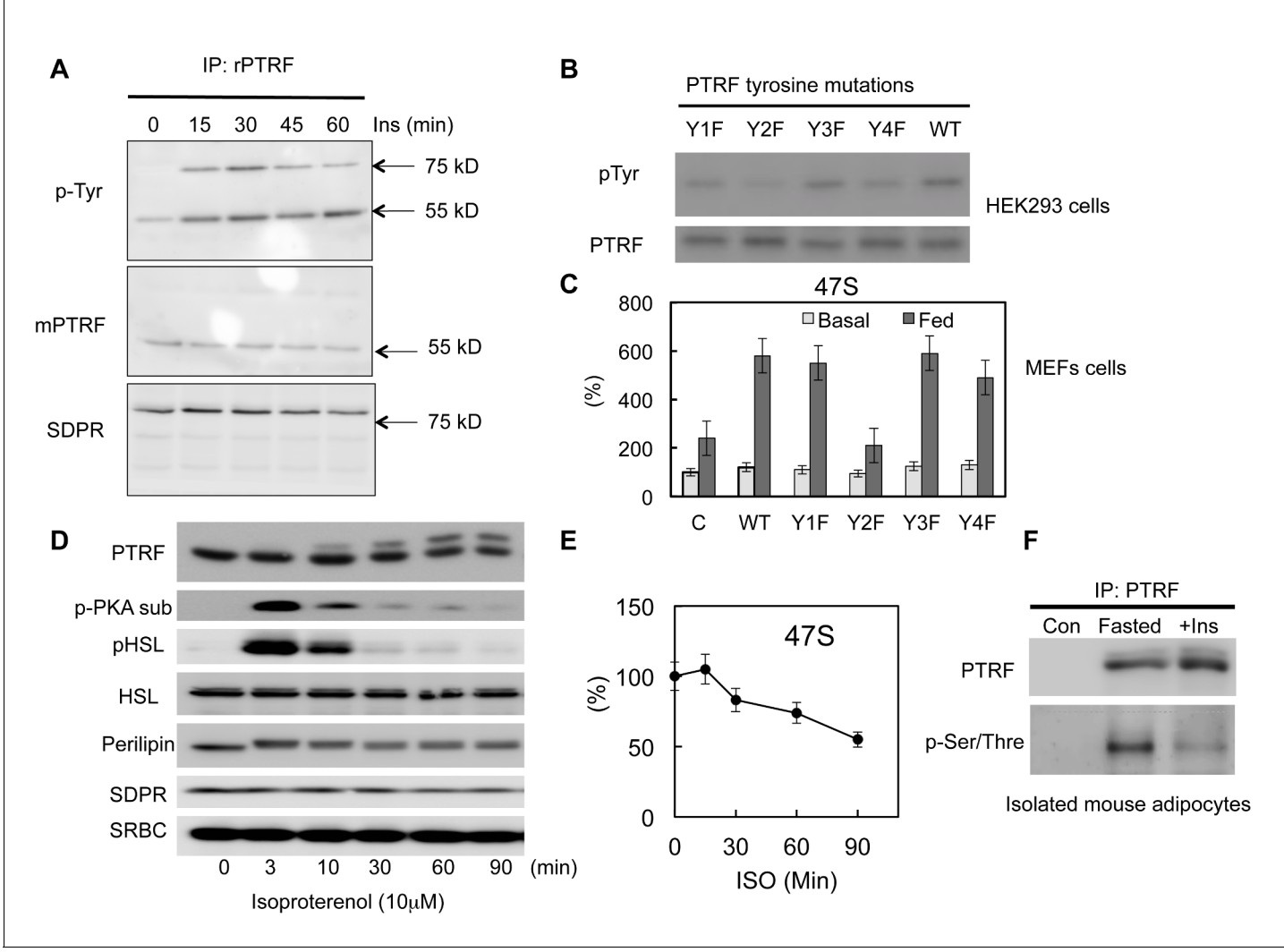

**Figure 5.** PTRF regulates ribosomal transcription through tyrosine and Ser/Thre phosphorylations. (A) Isolated mouse primary adipocytes were stimulated by insulin from 0–60 min. The whole cell lysates were immunoprecipitated by PTRF antibody followed by SDS-PAGE and blotted with indicated antibodies. (B) Whole cell lysates from wild type PTRF and various single tyrosine mutants (1: Y13F; 2: Y158F; 3: Y310F; 4: Y318F) transfected HEK293 cell were immunoprecipitated by PTRF antibody followed by SDS-PAGE and blotted with indicated antibodies. (C) Basal and nutrients/insulin stimulated 47S levels were measured from control, wild type PTRF and various single tyrosine mutants transfected PTRF null MEFs cells. (D) Whole cell lysates from 3T3-L1 adipocyte stimulated by isoproterenol (ISO) for 0–90 min were subject to SDS-PAGA and blotted with indicated antibodies. (E) 47S levels were measured from ISO stimulated 3T3-L1 adipocytes by RT-qPCR. (F) Isolated mouse primary adipocyte were nutrients starved (fasted) or stimulated by insulin with nutrients. The whole cell lysates were immunoprecipitated by PTRF antibody followed by SDS-PAGE and blotted with indicated antibodies.

## PTRF regulates ribosomal transcription through tyrosine and Ser/Thre phosphorylations

Previously we and others have shown PTRF can be tyrosine phosphorylated by EGF or insulin signaling pathway activations (*Guha et al., 2008*; *Humphrey et al., 2013*; *Pilch and Liu, 2011*; *Pilch et al., 2007*). However the functional relevance of this modification in a physiological context remains unknown. Consistent with previous results, we show here that insulin stimulates PTRF tyrosine phosphorylation (*Figure 5A*). Interestingly another caveolae protein in the cavin family, SDPR (Serum deprivation response protein, Cavin-2) co-immunoprecipitated with PTRF (*Bastiani et al., 2009*) also showed increased insulin-dependent tyrosine phosphorylation but it does not undergo nuclear translocation (*Figure 1C*). Next we investigated whether this modification plays any role in

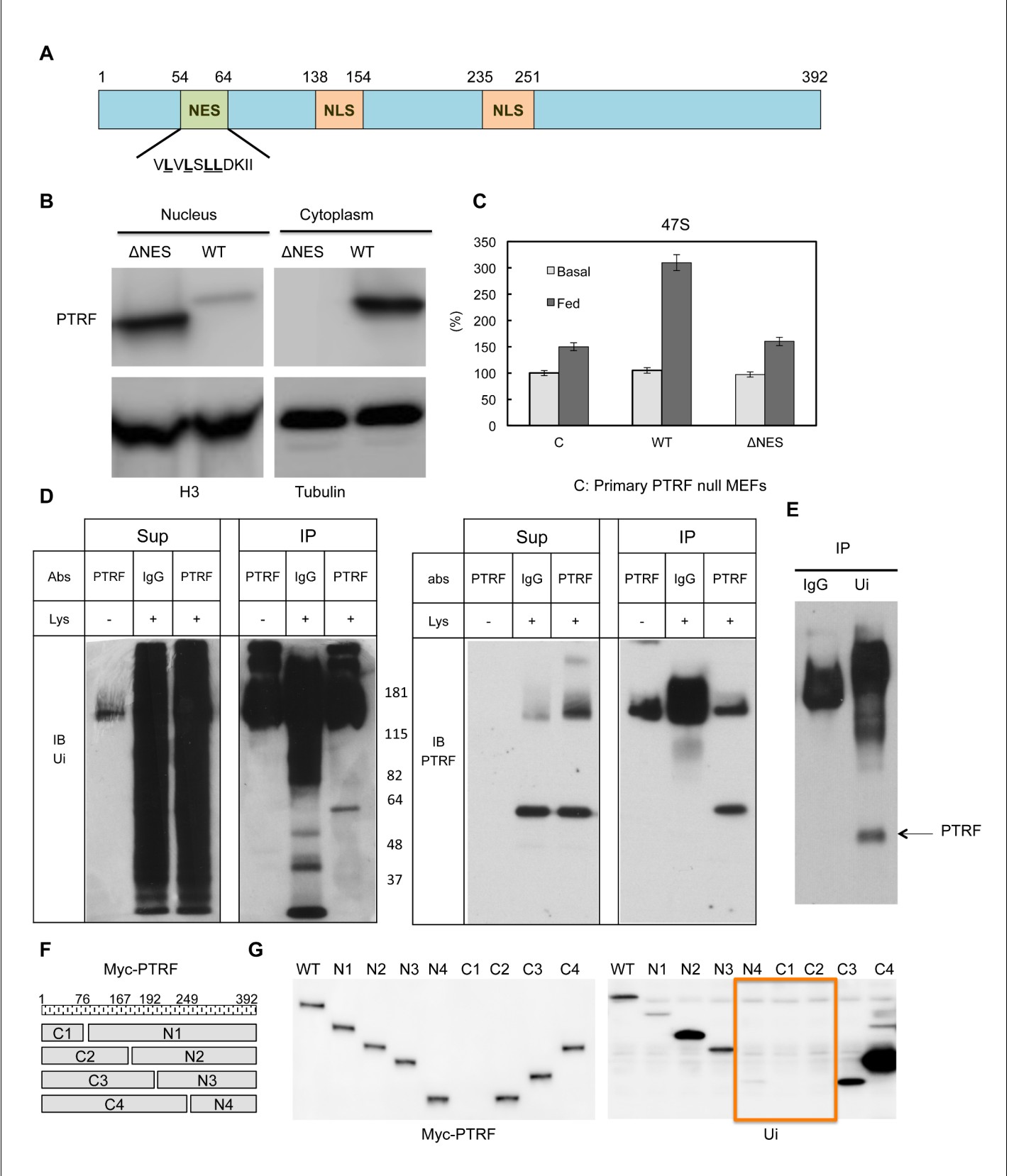

**Figure 6.** PTRF nuclear localization is regulated by its nuclear export signals and other post-translational modifications. (**A**) Schematic illustration of nuclear export signals (NES) and nuclear localization signal (NLS) motifs in mouse PTRF protein. (**B**) NES deleted PTRF constructs and wild type control

*Figure 6 continued on next page*

*Figure 6 continued*

were transfected to HEK293 cells. The nuclear and cytoplasmic fractions were separated on SDS-PAGE following by immunoblot using indicated antibodies. (C) Basal and nutrients/insulin-stimulated 47S levels were measured from empty vector controls (C), wild type (WT) and NES motif deleted PTRF cDNA constructs transfected PTRF null MEFs cells. (D and E) Total rat adipocyte lysate were immunoprecipitated by IgG, or PTRF or ubiquitin (Ui) followed by SDS-PAGE and blotted with indicated antibodies. (F) A scheme of 8 designed PTRF fragment mutants (C1-4, N1-4). (G) Whole cell lysates from Wild type HEK293 cells transfected by (WT) and 8 PTRF fragment mutants were immunoprecipitated by myc-tag magnetic beads followed by immuno-blots with indicated antibodies.

rDNA transcription regulation. Since PTRF has only 4 tyrosine sites, we substituted each of these with phenylalanine, Y158F showed a significant decrease in phosphotyrosine levels (*Figure 5B*). Accordingly, this mutant form (Y158F) shows a significant inhibitory effect on rDNA transcription activity when transfected into PTRF null MEF cells (*Figure 5C*), suggesting a critical role for this phosphorylation site in mediating the up-regulation of rDNA transcription activity in response to feeding medium.

PTRF has also been shown to be serine/threonine phosphorylated (*Aboulaich et al.,2004, 2011*). Upon PKA activation by isoproterenol (ISO) in adipocytes, PTRF shows a significant size shift on SDS/PAGE. Consistent with this we showed a slower PTRF band migration in SDS page following ISO stimulation (*Figure 5D*). Furthermore we showed this PKA pathway-dependent phosphorylation associated with a down-regulation of rDNA transcriptional activity measured by 47S rRNA levels (*Figure 5E*), suggesting a potential mechanism for negatively regulated ribosome transcription through PTRF Ser/Thre-phosphorylation under catabolic conditions. We also show this Ser/Thre-phosphorylation can be reversed by insulin (*Figure 5F*). These data support the hypothesis that PTRF is a key metabolism-dependent transcriptional efficiency regulator that can be both positively and negatively regulated through different signaling pathways, thus fulfilling its physiological functions in adipocytes under various nutrient and hormonal conditions.

## PTRF nuclear localization is regulated by its nuclear import and export sequences

Next we probed the potential mechanism(s) for PTRF translocation between different cellular localizations. PTRF has two putative nuclear localization signals (*Aboulaich et al., 2004, 2006*; *Kalderon et al., 1984*; *Wei et al., 2015*). Besides this, PTRF contains a protein sequence ([54]VLVLSLLDKII[64]), which is highly similar to nuclear export signals (NESs) (*Figure 6A*). The NES is a short amino acid sequence of hydrophobic residues in a protein that targets it for export from the cell nucleus to the cytoplasm through the nuclear pore complex by interacting with exportin, and the common NES sequence motif is 'LxxLxxLxL', where 'L' is a hydrophobic residue (often leucine) and 'x' is any other amino acid (*Bogerd et al., 1996*). To investigate if this NES signal plays any role in PTRF nuclear localization, we generated a mutant by deletion of this region and expressed it in HEK293 cells. As shown in *Figure 6B* this mutant was exclusive localized in the nucleus, probably due to the loss of nuclear export signal. When we transfected it into PTRF null primary MEFs cells, unlike wild type, this mutant cannot rescue 'feeding' medium 47S up-regulation under cell growth arrest condition (*Figure 6C*), indicating a nuclear export 'recycling' process is needed for maintaining PTRF ribosomal transcription regulatory activity.

Mouse PTRF has 392 amino acids and the predicted size is 43 kDa. However the band in SDS-PAGE always migrates at 55–60 kDa. By immunoprecipitation and immunoblotting, we showed that PTRF is mono-ubiquitinated (*Figure 6D and E*). Based on the size of 8 kDa for each ubiquitin protein, PTRF seems to be mono-ubiquitinated at two sites. Thus we expressed 8 PTRF truncation constructs (*Figure 6F*) in HEK293 cells and immunoprecipitated them with anti-myc antibody followed by western blot using an anti-ubiquitin antibody. As shown in *Figure 6G*, it seems that two regions (166-192aa and 193-249aa) are responsible for one mono-ubiquitination each. Protein monoubiquitination has recently emerged as an important posttranslational modification regulating transcription, endocytic vesicle trafficking, histone modification, and DNA repair (*Friedberg, 2006*; *Pemberton and Paschal, 2005*). For example, the addition of a mono-ubiquitin tag to p53 is sufficient for its nuclear export; p53 monoubiquitination may cooperatively interact with its nuclear export signal (NES) for cytoplasmic targeting (*Carter et al., 2007*). NLS can also be regulated by

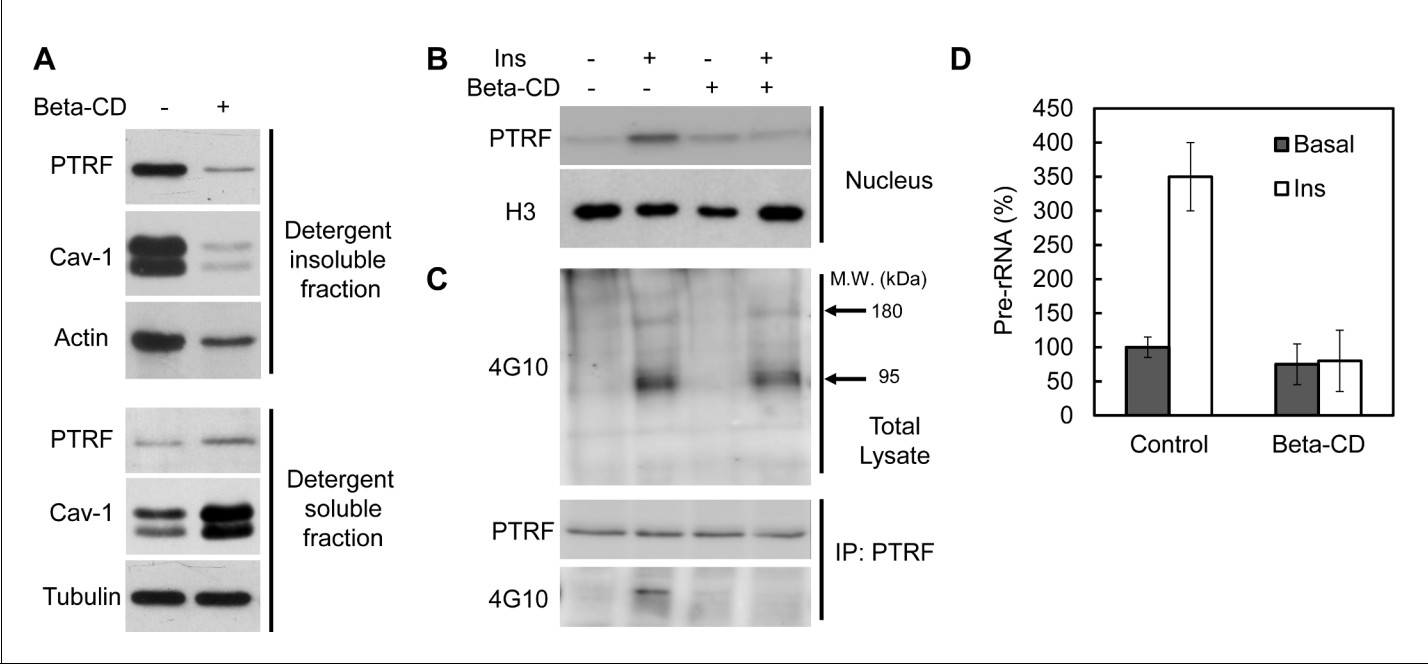

**Figure 7.** Intact caveolae are required for PTRF tyr-phosphorylation, nuclear translocation and ribosomal function. (**A**) Isolated mouse primary adipocytes treated with beta-cyclodextrin (Beta-CD) or not were fractionated into crude lipid rafts and non lipid rafts fractions. Lysate from both fractions were run on SDS-PAGE and blotted with indicated antibodies. (**B**) Nuclear fraction lysates from beta-CD (45-min) and insulin (30-min) treated or not adipocyte were run on SDS-PAGE and blotted with indicated antibodies, histone-H3:H3. (**C**) Whole cell lysates from **B** were separated by SDS-PAGE and blotted with phosphotyrosine antibody (4G10). Arrows indicate the size of 180 and 95 kDa, which possibly corresponds to IRS-1/2 and beta-subunit of insulin receptor respectively. The whole cell lysates were also immunoprecipitated by PTRF antibody followed by SDS-PAGE and blotted with indicated antibodies. (**D**) 47S levels of the samples from **B** were measured by RT-qPCR.

monoubiquitination signals. Studies have suggested that monoubiquitination can mask its NLS resulting in cytoplasmic retention (*Chen and Mallampalli, 2009*). Our data in *Figures 5* and *6*, taken together, suggest the localization of PTRF in caveolae or in the nucleus might be coordinated by monoubiquitination, NLS and NES motifs.

## Intact caveolae are required for PTRF tyr-phosphorylation, nuclear translocation and ribosomal function

To test if intact caveolae are required for PTRF ribosomal functions we treated mouse primary adipocyte with beta-methyl-cyclodextrin (beta-CD), which has been widely used to disrupt caveolae structure due to cholesterol depletion (*Wiesmann et al., 1975*; *Liu and Pilch, 2008*). As shown in *Figure 7A*, beta-CD treatment caused PTRF and caveolin-1 redistribution out of lipid raft fractions, consistent with previous reports. However insulin-stimulated PTRF nuclear translocation was completely blocked (*Figure 7B*), indicating the localization of PTRF in caveolae is critical for this translocation. Furthermore immunoblot analyses of tyrosine-phosphorylated proteins in cell lysates showed that PTRF tyrosine phosphorylation was largely inhibited although the insulin stimulated IRS-1/2 (180 kDa) and the beta-subunit of insulin receptor (95 kDa) (arrows) were not affected by beta-CD treatment (*Figure 7C*). PTRF mediated ribosomal transcription regulation was diminished as shown in *Figure 7D*. These results suggest intact caveolae structures provides a platform for insulin stimulated PTRF tyrosine phosphorylation, which is essential for its nuclear translocation and ribosomal function.

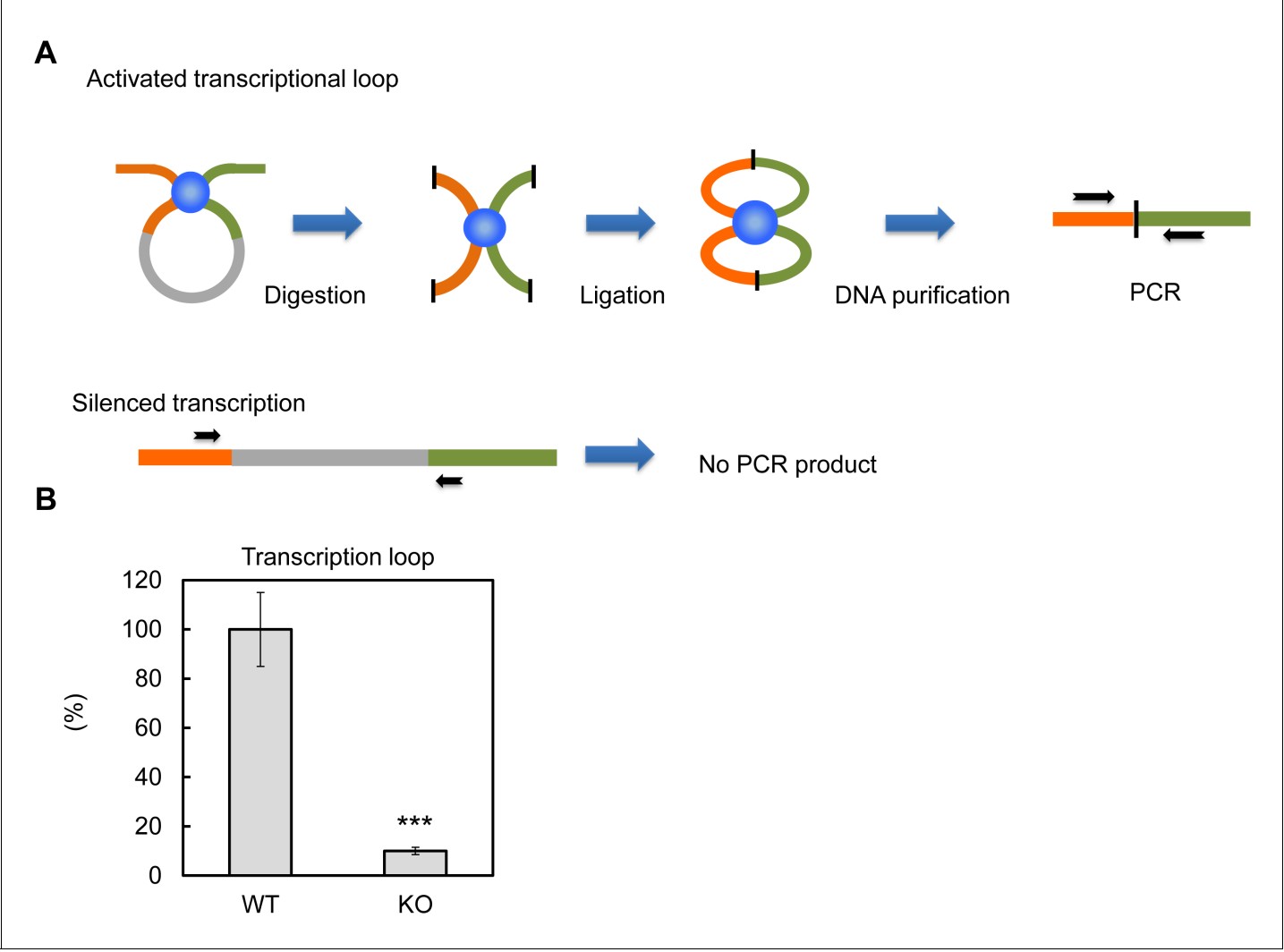

**Figure 8.** PTRF dependent active ribosomal transcription complex formation. (**A**) Schematic illustration of chromatin confirmation capture (3C) assay, details are described in Method. (**B**) Following 3C assay, the relative levels of active transcription were quantified by RT-qPCR using nutrition/insulin stimulated primary mouse adipocytes from wild type (WT) and PTRF null (KO) mice (n = 4). ***p<0.001; Student's test. Error bars indicate SD.

## PTRF plays a critical role in the formation of the ribosomal transcriptional 'loop'

Previous in vitro assays suggested PTRF up-regulates rDNA transcription rate through 'releasing' Pol-I from the transcription termination complex, which results in more efficient transcriptional re-initiation (*Jansa and Grummt, 1999*). The active rDNA transcription complex forms nucleotide loops by jointing initiation and termination regions together (*Grummt and Langst, 2013*). Here we tested if PTRF plays any role on this active loop formation. By using a chromosome conformation capture assay (3C assay) (*Nemeth et al., 2008*), we pulled down the transcriptional complex and ligated the DNA fragments followed by PCR using primers targeting the initiation and termination loci (*Figure 8A*). As shown in *Figure 8B* the PCR product was almost completely absent in PTRF null cells comparing to control WT cells, indicating the existence of a PTRF-dependent active rDNA transcription loop formation upon nutrient and insulin stimulation.

## Discussion

Publications on PTRF/Cavin-1 in the past 8 years have focused almost exclusively on its essential role as a structural component of caveolae, and these structures are especially abundant in fat cells (*Parton and del Pozo, 2013*). When PTRF/Cavin-1 is knocked out in mice or lost through mutation in humans, caveolae are absent and the metabolic consequences are severe, including muscular and lipodystrophies. The possible explanation(s) proposed for the adipocyte lipodystrophic phenotype include alterations in lipid transport and/or altered plasma membrane adaptations (*Pilch and Liu, 2011*), but these do not fully explain the progressive nature of the lipodystrophy observed in vivo. The transport/membrane defects would be present at birth presumably causing an immediate fat cell deficiency. On the other hand and given that the lipodystrophy of PTRF/Cavin-1 null mice and humans has a progressive severity, PTRF's eponymous function in rRNA synthesis does indeed provide a possible explanation whereby decreased rRNA transcription as a function of time would lead to adipocyte loss due to the inability to maintain adequate protein levels, and we present data here in support of this hypothesis.

It was recognized some time ago that a fraction of PTRF could be found in the nucleus of human fat cells consistent with it having two nuclear localization sequences, but the possible functional significance of this finding was not pursued (*Aboulaich, 2004*). Here we confirm that PTRF is present in the nucleus of fat cells, and indeed, it functions in these cells to promote rRNA transcription (*Figures 1–3*). The data of *Figures 1–3* derive from a comparison of WT and PTRF null primary mouse fat cells and gene edited cultured fat cells, and taken together, support the hypothesis that inefficient rRNA transcription underlies the lipodystrophy observed in mice and humans. Moreover, the absence of PTRF would be expected to lead to ribosomal stress, which in turn would lead to fat cell loss over time, and we find evidence for this process in fat cells from PTRF null mice (*Figure 4*).

The nuclear localization of PTRF is dynamic and regulated by insulin-dependent tyrosine phosphorylation under anabolic conditions and Ser/Thre phosphorylation under catabolic conditions (*Figure 5*). Mutation of tyr-158 to phe abrogates the regulation of rRNA transcription and prolonged exposure to the beta-adrenergic agent, isoproterenol, leads to ser/thre phosphorylation and diminution of 47S transcription, the catabolic condition (*Figure 5*). These results make physiological sense by adjusting potential ribosomal activity to the nutritional status of the cell. A further level of regulation may be achieved by ubiquitination and we have verified the necessity of the nuclear export signal in PTRF dynamics (*Figure 6*). Our data (*Figure 7*) support the idea that intact caveolae are required for PTRF tyrosine phosphorylation and nuclear function. Further conceptual support for this derives from the fact that the relevant kinases are very likely to be located at the plasma membrane but further experiments regarding this and the other PTRF modifications will be necessary to complete the picture.

Lastly for ribosome transcription, it's also proposed that 'loop' formation between initiation and termination complexes is essential for activation (*Denissov et al., 2011*; *Nemeth et al., 2008*; *Nemeth and Langst, 2011*; *Sander and Grummt, 1997*; *Shiue et al., 2009*). Our studies support that PTRF play an essential role in this loop formation (*Figure 8*).

We show that in 3T3-L1 pre-adipocytes, there is a high level of Pol-I expression in order to ensure the needs of rRNA syntheses by increasing the number of active transcription sites. This is closely correlated with pre-adipocyte proliferation. Upon adipocyte differentiation and consistent with previous results (*Li et al., 2006*) Pol-I levels were dramatically decreased (*Figure 3C*), whereas PTRF and TTF-1 levels were increased correspondingly, indicating ribosomal transcription in mature adipocyte became less Pol-I dependent, and more reliant on the regulation of transcriptional efficiency through the termination machinery. Our results support the notion that cells have developed at least these two different mechanisms to ensure proper control of ribosomal transcription in a cell type and physiological condition-specific manner. One mechanism is Pol-I dependent active transcriptional number regulation, which plays major role in proliferating cells. The other is termination factor, PTRF-dependent transcriptional rate regulation, which plays a critical role in mature differentiated cells in response to nutrient challenges. In fact, all cells must maintain a certain level of ribosomal activity to maintain protein expression upon their normal turnover, and nutritionally stressed cells such as adipocytes have a particular need for this activity. These cells are subject to repeated cycles of feeding and fasting, and indeed, we show in vitro (*Figure 3E*), the consequences of PTRF deficiency in this regard. We see no apparent differences in protein and fat content between WT and

PTRF null fat cells until we expose them to the metabolic stress of repeated fasting and refeeding cycles. The PTRF null cells show a dramatic reduction in total triglyceride, RNA and protein content which essentially recapitulates the lipodystrophy in vitro (*Figure 3E*) that is occurring in vivo in PTRF deficient mice and humans.

The PTRF null status obviously represents the extreme, but variations in PTRF expression would be predicted to have phenotypic consequences, in particular, in response to overnutrition, a ubiquitously prevalent condition in current times. What has been observed is that some individuals respond to over-nutrition by fat cell expansion requiring ribosomal activity to amass the additional proteins required to increase the size of fat cells and accommodate the excess nutrients as triglyceride, the so-called healthy obese individuals who lack obvious metabolic pathology. On the other hand, most people apparently fail to mount an adequate adipocyte adaptive response, and this results in fat spilling over to liver and muscle resulting in hepatosteatosis and type 2 diabetes. We hypothesize that the capacity of adaptational ribosomal DNA transcription may determine the maximal limit of adipocyte functional homeostasis and may coincide with these metabolic perturbations. In support of this idea, mice were exposed to conditions of normal, over and under nutrition from birth to weaning and then subjected to the same diet and followed for up to 112 days. The mice subject to undernourishment show an apparent inability for adipose tissue expansion that correlates with PTRF expression (*Kozak et al., 2010*). Together these data and ours suggest the PTRF-dependent ribosomal transcriptional response to metabolic challenges may be the early and direct causal molecular mechanism for the pathological developments of adipose tissue function in vivo, lipodystrophy in the worst case scenario.

## Materials and methods

### Animal
$Ptrf^{-/-}$ mice were created as described in (*Liu et al., 2008*). They were backcrossed for at least 8 generations with the C57 black lineage. The mice used in the present study were homozygous male $Ptrf^{-/-}$ and their wild-type (WT) littermates generated from breeding of $Ptrf^{+/-}$ mice. Animals were maintained in a pathogen-free animal facility at 21°C under a 12-hr light/12-hr dark cycle with access to a chow diet (CD, 2918; Harlan Teklad Global Diet, Madison, WI). For the preparation of isolated adipocytes, freshly harvested adipose tissue was digested by collagenase in Krebs-Ringer bicarbonate (KRB) buffer. Isolated rat adipocytes were prepared by the collagenase method as described in (*Liu et al., 2006*) from epididymal adipose tissues of Sprague-Dawley rats (from Charles-River, 170–220 g). All animal studies were performed in accordance with the guidelines and under approval of the Institutional Review Committee for the Animal Care and Use of Boston University.

### Cell culture
3T3-L1 fibroblasts culture and differentiation were previously described in (*Liu and Pilch, 2008*). PTRF null and control wild type primary MEFs cells used in all experiments were maintained within 3–5 passages. When they reached 90% confluence, they were transfected with the cDNA of interest by means of the X-tremeGENE HP reagent (Roche, Indianapolis, IN). HEK293 cells were cultured and transfected as describe in (*Liu and Pilch, 2008*).

### Generation of PTRF null 3T3-L1 stable cell lines by CRISPR/Cas9 genome editing
The PTRF null 3T3-L1 stable cell lines were generated using lentiviral plasmid based CRISPR/Cas9 genome editing, in which U6 promoter driven guide RNA, CMV promoter driven cas9 and puromycin selection marker were supplied by one single-vector (all-in-one). Four vectors, including three *mus musculus Ptrf* targets (K1: 5'-TCACGCTCCATATCGTTGAG-3'; K2: 5'-GTCAACGTGAAGACCG TGCG-3'; K3: 5'-GGTCAGCTGGATCTGGTCAA-3') and one non-targeting control were custom designed and made by Transomic Technologies (Huntsville, AL). Lentivirus were packing used third generation packing system. The stable cell lines were obtained after lentivirus transduction and puromycin selection. These cells were cultured and differentiated as described above.

## Reagents

Dexamethasone, 3-isobutylmethylxanthine, insulin, sodium fluoride, sodium orthovanadate, fetal bovine serum (Australian origin), benzamidine, and mouse immunoglobulin G (IgG) were purchased from Sigma (St. Louis, MO). LB base, ampicillin, kanamycin, aprotinin, leupeptin, and pepstatin A were obtained from American Bioanalytical (Natick, MA). Calf serum was purchased from Life Science (Cambridge, MA), and Dulbecco's modified Eagle's medium (DMEM) was from Mediatech (Herndon, VA). Transfection reagent and the pcDNA 3.1 expression vector were purchased from Life Science. A BCA protein assay kit was from Pierce. Protein A or G magnetic beads was from Santa Cruz Biotechnology (Santa Cruz, CA). Penicillin, streptomycin, and trypsin were purchased from Life Science.

## Antibodies and western blotting

Monoclonal antibodies recognizing PTRF (2F11), caveolin-1 (7C8), have been previously described (*Souto et al., 2003*; *Vinten et al., 2001*). The following antibodies were commercially acquired: anti-caveolin-1 was from BD Transduction Laboratories (San Jose, CA), anti-actin was from Sigma; anti-transferrin receptor was from Zymed Laboratories (South San Francisco, CA). Additional anti-PTRF antibodies were purchased from BD Transduction. Polyclonal rabbit anti-PTRF antibody was also produced against a peptide sequence at the C terminus of the protein (21st Century Biochemicals, Hopkinton, MA). Primary antibodies were detected in Western blots using secondary antibodies conjugated to horseradish peroxidase (Sigma) diluted 1:3000 and chemiluminescent substrate (Perkin-Elmer Life Sciences, Boston, MA), followed by detection by Fujifilm LAS-4000 Image Analyzer.

## Nucleus fractionation

Nucleus fraction was prepared according to sucrose centrifugation method (Nuclei Isolation Kit, 'Nuclei PURE Prep', Sigma, NUC201). This protocol incorporates centrifugation through a dense sucrose cushion to protect nuclei and strip away cytoplasmic contaminants.

## Co-immunoprecipitation

The whole cell/tissue lysates or soncicated nucleus fraction was solubilized with 1% Triton X-100. Insoluble material was removed by pelleting for 10 min in a microcentrifuge. Indicated antibodies and nonspecific mouse or rabbit IgGs were incubated with the supernatant 1 hr at 4°C, then 20–40 μl of protein A/G magnetic beads was added for 2 hr to overnight. The supernatant with unbound proteins was collected, and the beads were washed four times and eluted with SDS-PAGE loading buffer containing 2% SDS.

## Confocal microscopy

This protocol for isolated rat adipocyte and 3T3-L1 cultured adipocyte was performed as described by (*Liu et al., 2006*) and (*Liu and Pilch, 2008*).

## RNA extraction, cDNA synthesis and quantitative real-time PCR

Total RNA was isolated from indicated tissues or cells with TRIzol reagent (Life Science), and the cDNA was synthesized using Reverse Transcription System (Promega). Real-time PCR was performed with the ViiA7 detection system (Applied Biosystems) using Fast SYBR Green Master Mix (Applied Biosystems). Gene expression levels were presented relative to the wild type. The primer sequences are listed in supplementary file.

## Chromosome conformation capture analysis

Briefly, control and stimulated cells were fixed in 1% paraformaldehyde for 10 min at room temperature followed by adding glycine (0.6 M, final concentration) to quench the cross-linking reaction. Then cells were suspended in lysis buffer (10 mM Tris-HCl, pH 7.2, 10 mM NaCl, 0.2% NP-40 and protease inhibitor cocktail; Sigma) for 60 min on ice. The nuclei were pelleted, resuspended in nuclear buffer with 0.3% SDS and incubated for 1 hr at 37°C. Triton-X100 (1.8%) was added to sequester the SDS followed by the addition of high concentration BglII and BamHI (NEB) to digest the chromatin overnight at 37°C with gentle shaking. The restriction enzymes were inactivated by the addition of SDS to 1.6% and incubation at 70°C for 15 min. The reaction mixture was diluted

with 1x ligase buffer (NEB) and incubated for 1 hr at 37°C. Ligation of DNA was done using T4 ligase for overnight at 16°C. Following reversal of cross-linking and DNA extraction, the samples were subjected to qPCR quantification. The various primer combinations were tested for their amplification efficiency using a control template prepared by mixing mouse rDNA plasmids in equimolar amounts, followed by digestion with BglII and BamHI and subsequent ligation.

## Chromatin immunoprecipitation (ChIP) assays

ChIP assays were performed according to the manufacturer's protocol (MAGnify Chromatin Immuno-precipitation System, Thermo Fisher Scientific). Briefly antibodies were incubated with crosslinked chromatin overnight at 4 degree and collected with protein A magnetic beads. After reversal of the crosslink and digestion with proteinase K, DNA was extracted and amplified by PCR. PCR products were first visualized on ethidium bromide-stained agarose gels then subjected to real-time PCR for quantification.

## Statistics

All results are presented as mean ± SD. p values were calculated by unpaired Student's t-test. *p<0.05, **p<0.01, and ***p<0.001. P<0.05 was considered significant throughout. For cultured and primary isolated cells, all experiments were performed independently at least three times. Animal studies were from 4–6 animal per group.

## Acknowledgements

We thank Drs. J Williams and T Palmer for PTRF fragment DNA constructs. This work was supported by the grants NIH DK-30425 and DK-092942 to PFP. The authors declare no financial conflict of interest.

## Additional information

### Funding

| Funder | Grant reference number | Author |
|---|---|---|
| National Institute of Diabetes and Digestive and Kidney Diseases | DK30425 | Paul F Pilch |
| National Institute of Diabetes and Digestive and Kidney Diseases | DK092942 | Paul F Pilch |

The funders had no role in study design, data collection and interpretation, or the decision to submit the work for publication.

### Author contributions

LL, Conception and design, Acquisition of data, Analysis and interpretation of data, Drafting or revising the article, Contributed unpublished essential data or reagents; PFP, Conception and design, Analysis and interpretation of data, Drafting or revising the article

### Author ORCIDs

Libin Liu, http://orcid.org/0000-0001-5056-1517
Paul F Pilch, http://orcid.org/0000-0003-1997-0499

### Ethics

Animal experimentation: All animal studies were performed in accordance with the guidelines and under approval of the Institutional Review Committee for the Animal Care and Use of Boston University (protocol# 13974).

## Additional files

**Supplementary files**
• Supplementary file 1. List of primers.

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
