## [Decision Letter]

Thank you for submitting your article "PTRF/Cavin-1 Promotes Efficient Ribosomal RNA Transcription in Response to Metabolic Challenges" for consideration by *eLife*. Your article has been favorably evaluated by James Manley (Senior editor) and three reviewers, one of whom is a member of our Board of Reviewing Editors. One of the three reviewers involved in the assessment of your article has agreed to reveal his identity: David E James (Reviewer #3).

The reviewers have discussed the reviews with one another and the Reviewing Editor has drafted this decision to help you prepare a revised submission.

Summary:

This paper is of high interest and presents a novel concept for the field of adipose biology. The data provide novel insights into the functions and mechanisms of action of PTRF that include its nuclear translocation in response to insulin, the signals within its structure that mediate this re-localization and identification of specific phosphorylation sites that appear to be required for the activity of PTRF on directing rDNA transcription. This unexpected dual role of PTRF is quite remarkable and will of high interest to the field.

Essential revisions:

An important issue that should be further clarified is the relationship between the role of PTRF in caveolae and its suggested role in regulating ribosomal RNA. While data on the NES mutant PTRF is an important result, it is also possible that intact caveolae are required for the overall effect of PTRF on rRNA. Thus, a key experiment would be to analyze the role of PTRF in cells that are deficient in caveolae (caveolin 1 KO cells) so that a clear distinction can be made between PTRF function on rRNA, independent of its role in caveolae, versus a possible role in caveolae that affects this former function.

Related to this issue, do other caveolae proteins translocate to the nucleus and does the release of PTRF affect any functions of caveolae? Further related to this issue of the relationship between intact caveolae and PTRF: is it possible that PTRF from other pools in the cytoplasm is translocating to the nucleus? Overall, the revised manuscript will have to clarify these relationships between PTRF action on rRNA and the potential role of caveolae in this function.

---

## [Author Response]

*Essential revisions:*

An important issue that should be further clarified is the relationship between the role of PTRF in caveolae and its suggested role in regulating ribosomal RNA. While data on the NES mutant PTRF is an important result, it is also possible that intact caveolae are required for the overall effect of PTRF on rRNA. Thus, a key experiment would be to analyze the role of PTRF in cells that are deficient in caveolae (caveolin 1 KO cells) so that a clear distinction can be made between PTRF function on rRNA, independent of its role in caveolae, versus a possible role in caveolae that affects this former function.

We have conducted a new series of experiments to address this point. As shown in the new Figure 7, when caveolae integrity was disrupted by cholesterol depletion using methyl-β-cyclodextrin (see Liu and Pilch, JBC 2008;283(7):4314-22 and Breen et al., PLoS One. 2012;7(4):e34516.), there was essentially complete loss of caveolae and consequently, insulin-stimulated PTRF tyrosine phosphorylation, nuclear translocation and effects on ribosomal transcription. The total level of PTRF is unchanged in this protocol. These data are consistent with intact caveolae being necessary for PTRF phosphorylation and subsequent proper targeting to the nucleus, hence its function on rRNA transcription. A new section describing these results has been added to the revised manuscript. Although caveolin-1 null cells might have been an ideal model to test PTRF caveolae-independent function, we and others have shown that the expression levels of caveolin-1 and PTRF are always linked to one other in existing physiological models such as cav-1 or cavin-1/PTRF null mice. Thus there is essentially no PTRF in the *Cav1* null mouse (Liu and Pilch, unpublished; Hansen et al., Nat Commun. 2013; 4: 1831). Together with the other figures presented in this manuscript, we believe our data clearly show a distinction between PTRF function in the nucleus and in caveolae.

*Related to this issue, do other caveolae proteins translocate to the nucleus and does the release of PTRF affect any functions of caveolae? Further related to this issue of the relationship between intact caveolae and PTRF: is it possible that PTRF from other pools in the cytoplasm is translocating to the nucleus? Overall, the revised manuscript will have to clarify these relationships between PTRF action on rRNA and the potential role of caveolae in this function.*

New data were added to Figure 1 showing the other caveolae proteins, Cav-1, cavin-2 and -3, do not undergo nuclear localization or any stimulated translocation. Thus there is no overall movement of caveolae, just PTRF, which undergoes a specific translocation from plasma membrane caveolae to the nucleus in response to insulin. In the context of caveolae positive cells, PTRF translocation through cellular compartment(s) other than caveolae, the cytosol in particular, appears to be unlikely. First, the data presented in the new Figure 7 are consistent with PTRF localization in caveolae being necessary for translocation. In the context of PTRF in caveolae positive cells, we showed (Breen et al., PLoS1, 2012) by cell fractionation that the amount of this protein in the cytosol is very small (<5%) compared to that at present in caveolae. Moreover, these data may be the result of cell disruption, not intrinsic localization, as PTRF/cavin-1 dissociates from caveolae upon membrane and osmolarity stress (Sinha et al., Cell 2011, 144:402-413) both of which occur during cell disruption. Thus we think our data, including the new Figure 7, clarify how PTRF localizes to caveolae, probably exclusively, and then translocates to the nucleus upon its insulin-dependent tyrosine phosphorylation.